# ReAD: End-to-End Autonomous Driving Made Safer with Efficient Reinforcement Learning

## Abstract

End-to-end autonomous driving has received extensive attention due to its simplicity and scalability. However, they are usually trained by imitation learning and thus may suffer from limited behavioral diversity and distribution mismatch. To address this, we introduce a **Re**inforcement-learning-based end-to-end **A**utonomous **D**riving (**ReAD**) framework that enhances existing driving models through structured policy refinement. We first observe that models trained with imitation learning tend to plan trajectories by repeating previous training data rather than inferring the optimal. We propose to recalibrate them using lightweight reinforcement learning updates to avoid catastrophic forgetting while promoting high-reward behaviors. To resolve the inefficient training by composite rewards, we propose to decompose the reward into semantically aligned components. Each component provides a well-defined optimization objective to enable the policy to independently learn and balance distinct objectives. This leads to more efficient exploration, better credit assignment, and significantly improved convergence compared to using a single comprehensive reward. Experiments on both the open-loop nuScenes and closed-loop NavSim benchmarks demonstrate the effectiveness of ReAD to improve the planning performance of autonomous driving models. ReAD provides an efficient and effective pathway for reinforcement-learning-based optimization in safety-critical autonomous driving systems.

## 1 Introduction

Recent years have witnessed significant advances in autonomous driving (Wang et al., 2024c; Jiang et al., 2023; Liao et al., 2025a; Sun et al., 2025; Huang et al., 2024a; Li et al., 2024c; Huang et al., 2023; Shi et al., 2016). Data-driven deep networks have dramatically improved the accuracy of driving perception, including object detection (Zhang et al., 2022; Li et al., 2024c; Wang et al., 2025; Liang et al., 2022), segmentation (Strudel et al., 2021; Huang et al., 2024b; Zuo et al., 2025), and motion forecasting (Ettinger et al., 2021; Zheng et al., 2024). Still, the planning performance determines the quality of the driving policy and thus serves as a critical evaluation metric for autonomous agents. The planning module responsible for generating safe and comfortable trajectories remains a core and challenging component (Huang et al., 2021; Liao et al., 2025a).

While conventional autonomous driving systems are divided into different components (Yadav et al., 2024; Wu et al., 2024; Bai et al., 2022; Yin et al., 2021), the recent end-to-end models receive increasing attention due to their better scalability to more training data. They map raw sensor inputs directly to planning trajectories or control commands, demonstrating impressive performance through large-scale dataset training. However, most existing end-to-end driving models are trained with imitation learning from human demonstrations. This inherently limits the exploration of diverse driving strategies and is constrained by the performance of human experts. Although recent works like (Chen et al., 2024b; Liao et al., 2025a) have adopted multimodal paradigms for diverse trajectory generation, their action distributions remain skewed towards imitation rather than optimization due to their training methodologies. This leads to poor robustness to out-of-distribution scenarios and additionally suffers from error accumulation.

To address these, we propose a **Re**inforcement-learning-based end-to-end **A**utonomous **D**riving (**ReAD**) framework to enable driving models to effectively learn from reward feedback. We construct a coherent system for end-to-end autonomous driving based on reinforcement learning. It

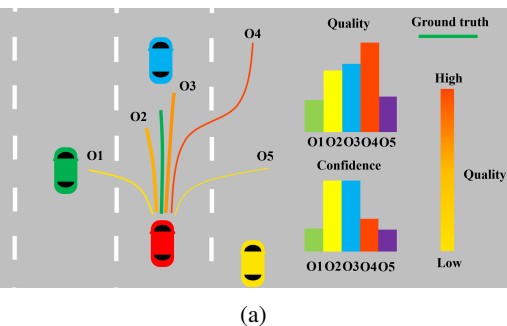 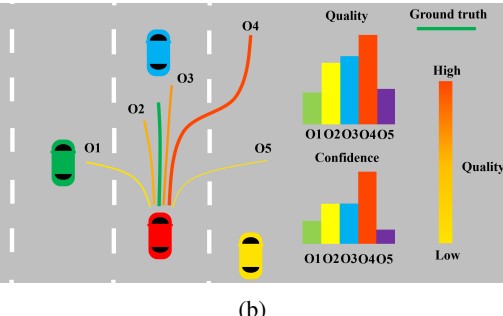

(a)            (b)

Figure 1. **Demonstration of confidence distribution over IL (Imitation-Learning) and RL (Reinforcement-Learning) trained agent.** Line-width of the trajectories represents the confidence score of the model towards them, while colors of the trajectories represent their quality evaluated by comprehensive reward metrics. (a) Models trained via imitation learning has a suboptimal confidence distribution over its action space, over-prioritizing trajectories that mimic expert demonstrations. (b) In reinforcement-learning based training, the distribution is recalibrated, encouraging deviation from expert-trajectories to explore high-reward patterns.

introduces a surrogate probability measure to resolve the computational challenge of probability density in continuous trajectory spaces. We also propose a decomposition strategy tailored for the GRPO (Group Relative Policy Optimization) reinforcement learning algorithm. We find that using a complex monolithic reward signal to construct a single GRPO policy gradient objective, which is the normal paradigm (Shao et al., 2024), tends to confuse the autonomous driving agent during RL training, as it obscures the specific causes of high rewards and makes it difficult for the agent to discern the underlying patterns. Instead, we decompose the complex reward signal into several semantically clear components, computing separate policy gradient losses for each of them and optimize a weighted sum of these losses. By doing so, the strategy provides the agent with a set of explicit targets, effectively guiding the policy to align them and achieve a harmonious optimization process.

Our method ReAD demonstrates favorable performance. When applied to the DiffusionDrive model on the NavSim benchmark, 2 epochs of reinforcement-based fine-tuning lead to an increase in the PDMScore (Dauner et al., 2024) from 87.7 to 88.8. This improvement is achieved with minimal additional tuning and without any modifications to the model architecture, suggesting the efficiency and effectiveness of ReAD. We further evaluate our method using open-loop metrics on the nuScenes dataset, with the DiffusionDrive-nusc branch as the baseline. The same brief training for 1 epoch significantly improves key planning metrics.The L2 error remains comparable (0.57 v.s., 0.57), and most notably, the collision rate is reduced by over 60%, achieving a new state-of-the-art result of 0.03% (from 0.08%). Extensive ablation studies verify that our reward decomposition strategy is pivotal to these gains, whereas using a single comprehensive reward alone leads to negligible improvement. Our method enhances end-to-end driving models by leveraging RL for efficient and targeted policy optimization, contributing to safer and more robust autonomous driving agents.

## 2 RELATED WORKS

**End-to-end Autonomous Driving.** End-to-end autonomous driving model directly maps raw sensor inputs to control commands or planned trajectories, seeking to replace conventional modular systems with a unified learning-based framework (Sun et al., 2025; Gao et al., 2024; Wang et al., 2024d; Liao et al., 2025b). UniAD (Shi et al., 2016) represents a key advance in this direction, introducing a query-based architecture that integrates multiple perception tasks under a single planning-aware objective. Subsequently, VAD (Jiang et al., 2023) proposed a vectorized scene representation with efficient query-based interactions, reducing computational overhead while retaining competitive planning performance. Despite these innovations, both models regress only a single trajectory. Many works adopt the same paradigm. (Chen et al., 2024c; Chitta et al., 2022; Li et al., 2024a;d) Such design overlooks the inherent multimodality and uncertainty in real-world driving scenarios. Addressing this limitation, VADv2 (Chen et al., 2024b) marks a paradigm change by incorporating a large vocabulary of 8192 anchor trajectories, which enable multi-mode motion predictions

and significantly improve the performance of the model. Hydra-mdp (Li et al., 2024b) further advances this direction. Diffusion-Drive (Liao et al., 2025a) applies powerful generative diffusion models to trajectory planning. By performing truncated denoising over a much smaller set of anchor trajectories, it generates highly diverse and rich trajectory proposals, substantially expanding the agent's action space and enabling more potential behavioral modes. Despite achieving a certain level of multimodality, however, Diffusion-Drive's planning process remains rooted in imitating expert trajectories during training. As a result, although the model possesses a rich action space, the probability distribution over this space tends to mimic expert behaviors rather than being explicitly optimized for driving performance. To this end, we propose READ,a reinforcement learning-based framework designed to fully unlock the multimodal capacity in end-to-end driving models.

**Reinforcement Learning for Autonomous Driving.** The integration of reinforcement learning into autonomous driving planning has been explored largely within the framework of Vision-Language-Action (VLA) models (Xu et al., 2024a; Chen et al., 2024a; Xu et al., 2024b; Wang et al., 2024a), where reinforcement learning is applied to refine decision-making based on semantic reasoning. AlphaDrive (Jiang et al., 2025) introduced a two-stage training strategy combining supervised fine-tuning with Group Relative Policy Optimization (GRPO), using multiple reward components tailored to driving objectives. Drive-R1 (Li et al., 2025) jointly optimized textual reasoning and trajectory prediction through a carefully designed reward function to reduce inconsistency. VLM-RL (Huang et al., 2025) automated reward shaping using VLMs, improving safety metrics, while OmniDrive (Wang et al., 2024b) integrated 3D spatial understanding and counterfactual reasoning to enhance robustness. However, few works have explored integrating reinforcement learning into end-to-end autonomous driving. This gap is noteworthy, as an end-to-end RL paradigm holds the potential to directly address key limitations of the prevailing multistage RL+VLA approaches. (For example, the reward misalignment and suboptimal policy updates arising from discrete reasoning stages) By mapping raw sensor inputs directly to control signals, end-to-end reinforcement learning is promising to achieve superior policy coherence, enhanced training efficiency, and more robust generalization in complex driving environments.

## 3 PROPOSED APPROACH

### 3.1 PRELIMINARIES

Reinforcement Learning (RL) is a machine learning paradigm where an agent learns to make decisions through trial and error by interacting with an environment. The goal is to maximize cumulative reward over time. At each time step, the agent observes a state $s_t$ and selects an action $a_t$ according to a policy $\pi$. In standard policy gradient algorithms, such as PPO (Schulman et al., 2017) and A2C (Mnih et al., 2016), a value network (critic) is typically employed to evaluate the quality of actions taken by the policy and compute the reward function, which guides the policy updates. While effective, training this value network can result in significant computational overhead, as it often doubles the number of parameters that need to be optimized. Group Relative Policy Optimization (GRPO) (Shao et al., 2024) is an RL algorithm designed to significantly reduce training costs by eliminating the need for a separate critic model.

The GRPO objective is defined as follows: For a given prompt $q$ (e.g., a state in a driving scenario), a group of $G$ outputs $o_1, o_2, \cdots, o_G$ is sampled from the old policy $\pi_{\theta_{old}}$. The new policy $\pi_\theta$ is then optimized by maximizing the objective:

$$J_{\text{GRPO}}(\theta) = \mathbb{E}_{q \sim P(Q), \{o_i\}_{i=1}^G \sim \pi_{\theta_{\text{old}}}(O|q)}$$

$$\left[ \frac{1}{G} \sum_{i=1}^G \min \left( \frac{\pi_\theta(o_i|q)}{\pi_{\theta_{\text{old}}}(o_i|q)} A_i, \text{clip} \left( \frac{\pi_\theta(o_i|q)}{\pi_{\theta_{\text{old}}}(o_i|q)}, 1 - \varepsilon, 1 + \varepsilon \right) A_i \right) \right] - \beta D_{\text{KL}}(\pi_\theta || \pi_{\text{ref}}), \quad (1)$$

where $\epsilon$ and $\beta$ are hyper-parameters. In the objective,the $D_{\text{KL}}(\pi_\theta || \pi_{\text{ref}})$ is the Kullback-Leibler divergence defined as:

$$D_{\text{KL}}(\pi_\theta || \pi_{\text{ref}}) = \left( \frac{\pi_\theta(o_i|q)}{\pi_{\text{ref}}(o_i|q)} - \log \frac{\pi_\theta(o_i|q)}{\pi_{\text{ref}}(o_i|q)} - 1 \right), \quad (2)$$

which acts as a regularizer that constrains the policy $\pi_\theta$ from deviating too far away from the reference model and consequently learns irrational or degenerate behaviors.

$A_i$ is the relative advantage given by:

$$A_i = \frac{r_i - \mu_{\mathcal{G}}}{\sigma_{\mathcal{G}}}, \quad \text{where } \mu_{\mathcal{G}} = \frac{1}{G} \sum_{j=1}^{G} r_j, \ \sigma_{\mathcal{G}} = \sqrt{\frac{1}{G} \sum_{j=1}^{G} (r_j - \mu_{\mathcal{G}})^2}. \tag{3}$$

Since the reward signal $r(\cdot)$ and the resulting advantage $A_i$ are required to be non-differentiable, GRPO does not rely on direct gradient-based optimization. Instead, the advantage scores provide a relative ranking of output quality within the group. By maximizing the objective, the probability distribution of the policy is adjusted to favor actions with higher advantages, thereby progressively improving performance.

### 3.2 REINFORCEMENT-LEARNING-BASED END-TO-END AUTONOMOUS DRIVING

Integrating reinforcement learning into end-to-end autonomous driving models poses unique challenges, primarily in relation to the continuity of the action space and the computation of the action probability $\pi_\theta(o|q)$ ,both central to policy gradient algorithms such as GRPO.

The application of policy gradient algorithms requires knowing the probability of generating each specific output $o_i$. In text-based domains (e.g, LLMs or VLM-based planners), the action space is naturally discrete, making probability computation straightforward. However, in end-to-end driving, the action space of the agent comprises high-dimensional continuous trajectories, rendering the probability density over this complex space computationally intractable. One common approach to overcome this is to discretize the continuous action space using a fixed set of trajectories, following the paradigm of VADv2 (Chen et al., 2024b). However, such discretization severely constrains behavioral expressivity, imposes considerable computational overhead, and compromises the generative continuity essential for nuanced trajectory planning.

We argue that policy optimization does not require the exact accurate probability density distribution over the continuous and complex trajectory space. The primary focus for $\pi_\theta$ lies not in precisely modeling the true action-space distribution, but rather in its ability to faithfully represent the preferences of the model and adapt effectively through training. Thus, a surrogate probability measure suffices if it is an optimizable confidence proxy and a differentiable output of the model.

**Optimizable Confidence Proxy**. The surrogate measure should be derived by carefully tracing the trajectory generation process of the agent, incorporating all stochastic stages while omitting deterministic components. Crucially, the probability distributions of these stochastic variables must be learnable and context-dependent—not based on fixed priors. This allows the measure to faithfully represent the model's confidence or preference for a specific trajectory output.

**Differentiable Variable**. The probability computation must be a differentiable output of the model and most crucially, must engage a substantial portion of the model's parameters. This facilitates extensive propagation of gradients throughout the network, enabling a comprehensive and effective policy re-calibration.

Formally, let the generation of a specific trajectory $o$ in an end-to-end model involve $M$ stochastic stages, each yielding a confidence logit $l_i$ for $i = 1, \ldots, M$. The overall surrogate log-probability is defined as:

$$\log \pi_\theta(o \mid q) = \sum_{i=1}^{M} \log \sigma(l_i), \tag{4}$$

where $\sigma$ denotes the softmax function (or an appropriate transformation to a probability value, as required per stage).

This surrogate probability measure provides the foundation for applying GRPO to end-to-end driving models. Nevertheless, the existence of optimizable probability distributions does not alone suffice for effective RL fine-tuning. It is equally critical that the base-model inherently supports multimodality, rather than merely producing minor variations of similar outputs.

Without meaningful diversity and multimodality, the trajectories sampled from the model would be nearly identical, rendering relative advantage comparisons in GRPO ineffective and failing to

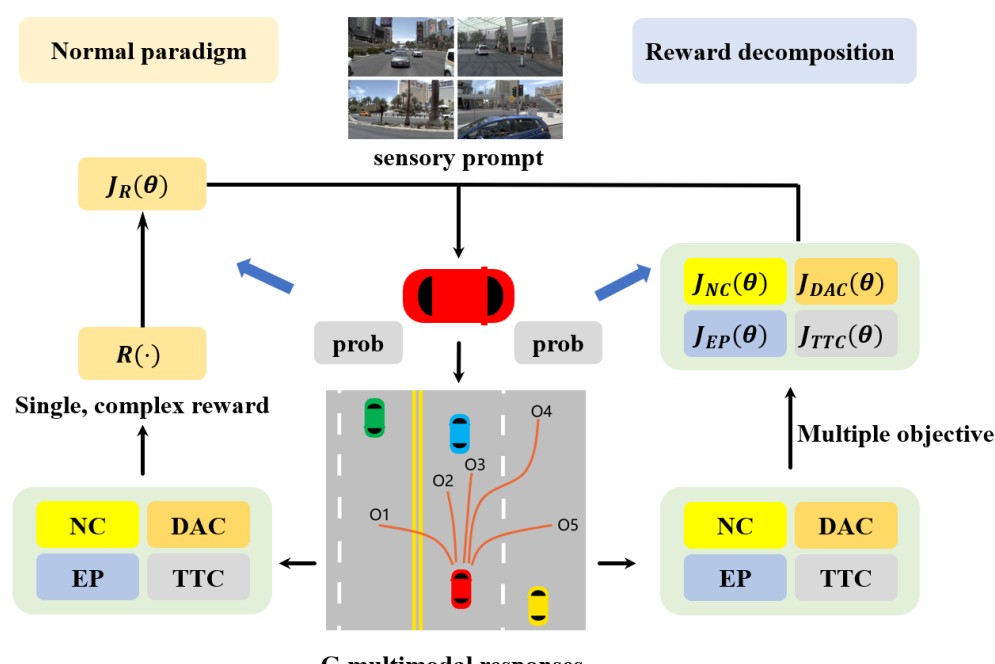

Figure 2. **Comparative demonstration of reward decomposition.** Comparison between normal paradigm of group-relative-policy optimization algorithm and our reward decomposition strategy provide a useful learning signal. On the other hand, for models already endowed with with a diverse action space, reinforcement learning can induce substantial performance gains through subtle adjustments in probability distribution.

### 3.3 REWARD DECOMPOSITION STRATEGY

The selection of reward signals is also a significant step in reinforcement learning.Since the action space of end-to-end autonomous driving agents is composed of trajectories, we need to comprehensively evaluate the quality of each sampled trajectory. A representative and widely-used metric for such evaluation is the PDMScore from the NavSim(Dauner et al., 2024) benchmark, which combines safety (**NC**: no-at-fault-collision, **DAC**: drivable-area-compliance),efficiency (**EP**:ego-progress),and comfort (**Comf**) metrics.

Our initial approach utilized the unified PDMScore as the sole reward signal $r_i = \text{PDMScore}(o_i)$ for each trajectory $o_i$. We computed a single advantage $A_i$ and policy gradient loss $\mathcal{L}_{\text{GRPO}}$ according to Equation (1). However, this strategy resulted in negligible improvement. We attribute this failure to the composite nature of the PDMScore: although it comprehensively reflects overall driving quality, its complex formulation obscures the underlying patterns that lead to high rewards, making it difficult for the policy to discern clear optimization directions during training.

In contrast, each component of the PDMScore provides semantically explicit and interpretable optimization guidance. To validate this insight, we conducted an experiment using only the EP score as the reward signal. The results (detailed in Section 4) demonstrated a remarkable improvement in ego progress. However, this single-objective optimization caused the agent to adopt overly aggressive behaviors, significantly degrading other metrics—particularly No-at-fault Collision (NC). This trade-off highlights a fundamental challenge: how to maintain clear and actionable optimization signals while holistically improving safety and efficiency.

Crucially, we observe that the various components of the PDMScore—though measuring distinct aspects—are not inherently competing or incompatible. Improvement in one metric (e.g., collision avoidance) need not come at the expense of another (e.g., progress). We therefore propose a reward decomposition strategy that preserves semantic clarity while enabling multi-objective optimization. We decompose the PDMScore into independent reward components:

$$r^{\text{NC}} = \text{NC}(o_i), r^{\text{DAC}} = \text{DAC}(o_i), r^{\text{TTC}} = \text{TTC}(o_i), r^{\text{EP}} = \text{EP}(o_i), r^{\text{Comf}} = \text{Comf}(o_i), \quad (5)$$

Figure 3. **Overall pipeline of ReAD.** For each prompt, the agent generates a group of G multimodal trajectories. ReAD then evaluates each of them using decomposed, semantically clear reward signals and computes multiple gradient policy losses with the surrogate probability measure. The agent then optimizes and balances the distinct objectives.

For each reward component $m \in \{\text{NC}, \text{DAC}, \text{TTC}, \text{EP}, \text{Comf}\}$, we compute a separate advantage $A_i^m$ using group-relative normalization:

$$A_i^m = \frac{r_i^m - \mu_G^m}{\sigma_G^m}, \quad \text{where} \quad \mu_G^m = \frac{1}{G}\sum_{j=1}^{G} r_j^m, \quad \sigma_G^m = \sqrt{\frac{1}{G}\sum_{j=1}^{G}(r_j^m - \mu_G^m)^2}, \quad (6)$$

We then derive a corresponding GRPO policy gradient loss for each component:

$$\mathcal{L}_{\text{GRPO}}^m = -\frac{1}{G}\sum_{i=1}^{G}\left[\min\left(\frac{\pi_\theta(o_i|q)}{\pi_{\theta_{\text{old}}}(o_i|q)}A_i^m, \text{clip}\left(\frac{\pi_\theta(o_i|q)}{\pi_{\theta_{\text{old}}}(o_i|q)}, 1-\epsilon, 1+\epsilon\right)A_i^m\right)\right] + \beta \cdot D_{\text{KL}}(\pi_\theta||\pi_{\text{ref}}),$$

$$(7)$$

The total optimization objective becomes a weighted sum of these individual losses:

$$\mathcal{L}\text{total} = \sum_m w_m \cdot \mathcal{L}_{\text{GRPO}}^m. \quad (8)$$

where $w_m > 0$ are weighting coefficients that balance the emphasis among different objectives.

This approach provides the agent with multiple distinct and interpretable learning signals, each guiding the policy toward a specific aspect of driving behavior. By adjusting $w_m$, we can explicitly control the optimization direction—emphasizing safety ($w_{\text{NC}}, w_{\text{DAC}}, w_{\text{TTC}}$), efficiency ($w_{\text{EP}}$), or comfort ($w_C$). As demonstrated in Section 4, this decomposition enables rapid and significant improvement in overall driving performance with minimal fine-tuning, achieving a better balance between all metrics compared to using a monolithic reward.

## 4  EXPERIMENTS

### 4.1  DATASETS

**NavSim.** NavSim is a high-fidelity simulation platform designed for benchmarking autonomous driving planning systems. It supports comprehensive closed-loop non-reactive evaluation with diverse urban scenarios and dynamic agent behaviors. We employ its PDMScore (as introduced in Section 3.3) for holistic assessment of driving performance across safety, progress, and comfort.

**nuScenes.** The nuScenes dataset is previously widely adopted for open-loop evaluation of end-to-end autonomous driving models. Although methods such as Ego-MLP(Li et al., 2024d) achieve strong performance on nuScenes using only ego-state information—suggesting limitations in the

Table 1: **Loss weights configuration** for ReAD training on NavSim.

| Loss Category | Loss Component | Weight |
|---|---|---|
| Policy Losses | Collision ($\mathcal{L}_{NC}$) | 5.0 |
| | Drivable Area ($\mathcal{L}_{DAC}$) | 4.0 |
| | Time-to-Collision ($\mathcal{L}_{TTC}$) | 7.0 |
| | Ego Progress ($\mathcal{L}_{EP}$) | 1.6 |
| | Comfort ($\mathcal{L}_{Comf}$) | 0.8 |
| Trajectory Losses | Total Trajectory | 4.0 |
| | Trajectory Classification | 2.0 |
| | Trajectory Regression | 1.6 |
| Perception Losses | Diffusion ($\mathcal{L}_{diff}$) | 0.20 |
| | Agent Classification ($\mathcal{L}_{cls}$) | 0.10 |
| | Agent Bounding Box ($\mathcal{L}_{box}$) | 0.01 |
| | BEV Semantic ($\mathcal{L}_{bev}$) | 0.14 |
| Other | KL Divergence ($\beta$) | 0.02 |

Table 2: **Comparison on planning-oriented NAVSIM navtest split.**

| Method | NC↑ | DAC↑ | EP↑ | TTC↑ | Comf.↑ | PDMS↑ |
|---|---|---|---|---|---|---|
| UniAD | 97.8 | 91.9 | 92.9 | 100.0 | 78.8 | 83.4 |
| Transfuser | 97.7 | 92.8 | 92.8 | 100.0 | 79.2 | 84.0 |
| VADv2-V8192 | 97.2 | 89.1 | 76.0 | 91.6 | 100.0 | 80.9 |
| Hydra-MDP-V8192 | 97.9 | 91.7 | 77.6 | 92.9 | 100.0 | 83.0 |
| Hydra-MDP-V8192-W-EP | 98.3 | 96.0 | 78.7 | 94.6 | 100.0 | 86.5 |
| DiffusionDrive | 98.1 | 96.2 | 81.8 | 94.5 | 100.0 | 87.7 |
| ReAD | **98.3** | **96.9** | **83.4** | 94.5 | 99.8 | **88.8** |

open-loop evaluation setup—they predominantly reduce L2 error by learning simplistic behaviors like straight-line driving or stopping, while still exhibiting high collision rates. Thus, within this framework, the collision rate remains a meaningful indicator of planning reliability.

## 4.2 IMPLEMENTATION DETAILS

**NavSim.** We fine-tuned the DiffusionDrive baseline using the ReAD framework with the GRPO algorithm. The entire model serves as the current policy model $\pi_\theta$, while the initial frozen model acts as the reference policy model $\pi_{\text{ref}}$. The total loss combines our decomposed-reward GRPO-policy-gradient losses with the original losses utilized in DiffusionDrive. Specifically,we retain the trajectory loss but reduce the weight of the perception losses to 10% to prioritize planning-related learning. The specific loss weights are detailed in Table 1.

The model was trained for 2 epochs (approximately 640 steps) on the `navtrain` split using a learning rate of $4.5 \times 10^{-6}$ and a batch size of 256. All experiments were conducted on 4 NVIDIA RTX 4090 GPUs with a random seed of 0. Model checkpoints were saved at 120-step intervals, and the results reported in the main paper correspond to the checkpoint at step 480, which achieved optimal performance on the validation set.

**nuScenes.** We also evaluated our ReAD framework on the DiffusionDrive-nusc branch. In this setting, we froze the perception layers and trained only the planning head of the current policy $\pi_\theta$, using the frozen initial model as the reference. We construct GRPO policy gradient loss for both the L2 loss and the collision rate metrics. The regression and collision-rate policy gradient loss weights are set respectively at 2.0 and 1.0,and the KL-divergence $\beta$ is set at 0.05. Training was conducted for 1 epoch (approximately 500 steps) using a learning rate of $2 \times 10^{-5}$ on 8 NVIDIA RTX 4090 GPUs. During training, checkpoints were saved at 100-step intervals. The results reported in this paper are

Table 3: **Open-loop evaluation results on the nuScenes dataset.**

| Method | L2 (m) ↓ | | | | Collision Rate (%) ↓ | | | |
|---|---|---|---|---|---|---|---|---|
| | 1s | 2s | 3s | Avg. | 1s | 2s | 3s | Avg. |
| VAD | 0.41 | 0.70 | 1.05 | 0.72 | 0.07 | 0.17 | 0.41 | 0.22 |
| UniAD | 0.45 | 0.70 | 1.04 | 0.73 | 0.62 | 0.58 | 0.63 | 0.61 |
| SparseDrive | 0.29 | 0.58 | 0.96 | 0.61 | 0.01 | 0.05 | 0.18 | 0.08 |
| DiffusionDrive | 0.27 | 0.54 | 0.90 | 0.57 | 0.03 | 0.05 | 0.16 | 0.08 |
| ReAD | 0.27 | 0.55 | 0.90 | 0.57 | **0.00** | **0.01** | **0.08** | **0.03** |

Table 4: **Ablation study on loss weights.**

| Experiments | GRPO Loss Weights | | | | | | Performance Metrics (NavSim) | | | | | |
|---|---|---|---|---|---|---|---|---|---|---|---|---|
| | $w_{NC}$ | $w_{DAC}$ | $w_{EP}$ | $w_{TTC}$ | $w_{COM}$ | $w_{PDMS}$ | NC ↑ | DAC ↑ | EP ↑ | TTC↑ | Comf↑ | PDMS ↑ |
| DiffusionDrive | – | – | – | – | – | – | 98.1 | 96.2 | 81.8 | 94.5 | 100.0 | 87.7 |
| ReAD | 5.0 | 4.0 | 1.6 | 7.0 | 1.0 | 0.0 | **98.3** | **96.9** | **83.4** | 94.5 | 99.8 | **88.8** |
| Experiment 1 | 0.0 | 0.0 | 0.0 | 0.0 | 0.0 | 10.0 | 98.1 | 96.2 | 82.0 | 94.4 | 99.9 | 87.7 |
| Experiment 2 | 0.0 | 0.0 | 10.0 | 0.0 | 0.0 | 0.0 | 94.3 | 95.5 | 82.7 | 84.0 | 93.5 | 81.8 |
| Experiment 3 | 5.0 | 4.0 | 1.6 | 8.0 | 1.0 | 0.0 | 98.3 | 96.8 | 83.1 | 94.8 | 99.7 | 88.7 |
| Experiment 4 | 5.0 | 4.0 | 2.0 | 7.0 | 1.0 | 0.0 | 98.2 | 96.8 | 83.4 | 94.4 | 99.7 | 88.7 |

achieved by the fourth or fifth saved checkpoint, both of which yield comparable performance on the evaluation metrics.

## 4.3 MAIN RESULTS

**NavSim.** On the NavSim benchmark, ReAD demonstrates a substantial improvement in holistic driving performance over the strong DiffusionDrive baseline. As shown in Table 2, after only 2 epochs of fine-tuning, ReAD elevates the overall PDMScore from 87.7 to **88.8**. Notably, READ achieves significant gains in Ego Progress (EP), improving from 81.8 to 83.4, while also enhancing safety-related metrics such as No-Collision (NC) and Drivable Area Compliance (DAC). These results validate that the lightweight reinforcement learning fine-tuning of ReAD effectively recalibrates the trajectory distribution toward higher-reward behaviors without compromising other aspects of driving quality.

**nuScenes.** Our open-loop evaluation on the nuScenes dataset highlights a dramatic improvement in safety. As summarized in Table 3, ReAD reduces the average collision rate by over 60% (from 0.08% to 0.03%) after just 1 epoch of training, achieving a new **state-of-the-art** result in collision avoidance. This significant safety enhancement is achieved while maintaining a comparable L2 error, underscoring the ability of ReAD to learn substantially safer policies without compromising trajectory accuracy.

## 4.4 ABLATION STUDY

**GRPO-Gradient-Policy-Loss Weight.** We conduct a detailed ablation study to evaluate the impact of different reward decomposition strategies and their corresponding loss weights on driving performance. The results, summarized in Table 4, demonstrate the critical role of our proposed reward decomposition.

In Experiment 1, we replace the decomposed reward with a single monolithic PDMScore signal ($w_{\text{PDMS}} = 10.0$). The results show negligible improvement over the baseline, confirming that a composite reward obscures the underlying patterns that lead to high rewards, thereby hindering effective policy exploration.

In Experiment 2, we use only the Ego Progress (EP) reward ($w_{\text{EP}} = 10.0$) to construct the policy gradient loss. While this leads to increased progress (EP rises from 81.8 to 82.7), it causes severe

Table 5: **Ablation study on anchor trajectory configurations.** Configuration notation: X/Y indicates training with X anchors and evaluation with Y anchors. The baseline uses 20 anchors for both training and evaluation (Liao et al., 2025a).

| Configuration | NC↑ | DAC↑ | EP↑ | TTC↑ | Comf.↑ | PDMS↑ |
|---|---|---|---|---|---|---|
| 20/20 (Baseline) | 98.1 | 96.2 | 81.8 | 94.5 | **100.0** | 87.7 |
| 20/20 (RL fine-tuned) | 98.3 | 96.5 | 82.5 | **94.8** | 99.9 | 88.4 |
| 20/40 (RL fine-tuned) | 98.3 | 96.6 | 82.6 | **94.8** | 99.8 | 88.4 |
| 40/20 (RL fine-tuned) | **98.4** | 96.6 | 83.1 | 94.7 | 99.8 | 88.6 |
| 40/40 (RL fine-tuned) | 98.3 | **96.9** | **83.4** | 94.5 | 99.8 | **88.8** |

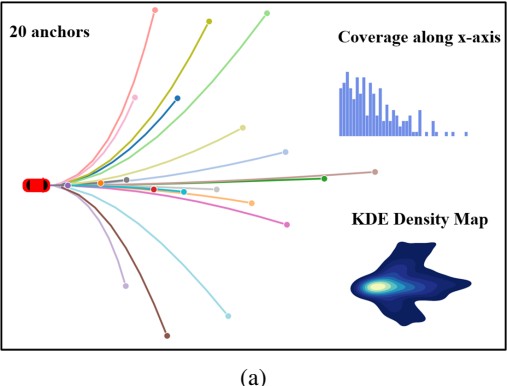 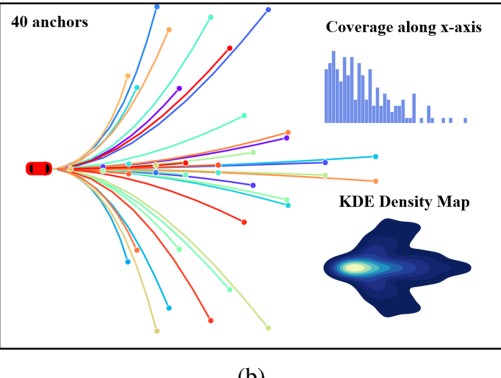

(a)                  (b)

Figure 4. **Visualization of anchor trajectories.** Forty anchor trajectories cover a more diverse action space and better facilitate multimodal exploration in reinforcement-learning based training.

degradation in safety metrics (NC drops to 94.3 and TTC to 84.0), indicating that single-objective optimization induces overly aggressive behavior.

Experiments 3 and 4 utilize multiple semantically clear reward signals to construct policy gradient loss. Both of the experiments show substantial overall improvement, validating that our decomposition strategy provides clear and complementary learning signals. Small adjustments in weights (e.g., $w_{TTC}$ or $w_{EP}$) lead to nuanced changes in agent behavior, balancing safety and efficiency without destructive trade-offs. Our final weight configuration achieves the best overall performance.

**Number of Anchor-Trajectories.** The structure of DiffusionDrive allows flexible use of anchor trajectories during training and inference. We ablate this flexibility to analyze its interaction with RL fine-tuning. As shown in Table 5, increasing the number of anchors enhances performance.

Using 20 anchors for both training and inference (20/20) already brings improvement over the baseline after RL fine-tuning (PDMS: 87.7 → 88.4). Using 40 anchors during training further boosts performance (PDMS: 88.8).Even infer with only 20 trajectories, the model still achieves the PDMS of 88.6, indicating that training with more diverse behavioral modes enables better policy recalibration. The best result is achieved with 40 anchors during both training and inference (40/40, PDMS: 88.8), confirming that a richer action space facilitates more effective exploration and probability redistribution during RL training.

## 5 CONCLUSION

In conclusion, this work introduces the ReAD framework, which explores the integration of reinforcement learning into end-to-end autonomous driving by addressing main challenges of policy optimization in a continuous trajectory space. The proposed method, featuring a surrogate probability measure to enable gradient-based updates and a reward decomposition strategy to provide balanced learning signals, offers an efficient fine-tuning pathway for pre-trained driving models. This work helps bridge the gap in applying RL to end-to-end driving systems, demonstrating a feasible direction for enhancing planning performance and safety with minimal computational overhead.

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

# A  APPENDIX

## A.1  DETAILS IN COMPUTATION OF SURROGATE PROBABILITY MEASURE

The integration of DiffusionDrive into ReAD framework requires the computation of a surrogate probability measure $\pi_\theta(o|q)$ for any generated trajectory $o$ given context $q$. We derive this by formalizing the trajectory generation pipeline of DiffusionDrive and analyzing its stochastic components.

DiffusionDrive generates multimodal trajectory proposals through a truncated diffusion process operating on a set of $K$ anchor trajectories $\mathcal{A} = \{\mathbf{a}_k\}_{k=1}^K$, where $\mathbf{a}_k \in \mathbb{R}^{T \times 3}$ ($T$ is the number of future timesteps). The complete trajectory generation procedure for a given scene context $\mathbf{q}$ (aggregating BEV features, agent queries, and ego status) can be formalized as:

$$\text{For each anchor } k = 1, \ldots, K, \ t \sim \mathcal{U}(1, T), \ \boldsymbol{\epsilon}_k \sim \mathcal{N}(\mathbf{0}, \mathbf{I}):$$
$$\tilde{\mathbf{a}}_k^{(t)} = \sqrt{\bar{\alpha}_t}\mathbf{a}_k + \sqrt{1 - \bar{\alpha}_t}\boldsymbol{\epsilon}_k; \ \mathbf{o}_k^{\text{reg}}, l_k = f_\theta(\tilde{\mathbf{a}}_k^{(t)}, \mathbf{q}); P(k|\mathbf{q}) = \frac{\exp(l_k)}{\sum_j \exp(l_j)}. \tag{9}$$

where $\bar{\alpha}_t$ is the noise schedule coefficient at timestep $t$, $\mathcal{U}$ denotes uniform distribution, and $f_\theta$ represents the decoder network that simultaneously predicts the denoised trajectory $\mathbf{o}_k^{\text{reg}}$ and the mode selection logit $l_k$ from the noise-trajectory feature.

We systematically examine the generation pipeline to identify optimizable stochastic elements:

- *Initial Noise Sampling*: The noise vectors $\boldsymbol{\epsilon}_k$ are sampled from a fixed isotropic Gaussian distribution $\mathcal{N}(\mathbf{0}, \mathbf{I})$. While this introduces stochasticity, the probability density $p(\boldsymbol{\epsilon}_k) = (2\pi)^{-d/2} \exp(-\frac{1}{2}\|\boldsymbol{\epsilon}_k\|^2)$ is a *fixed prior* that cannot be optimized through gradient-based learning.

- *Timestep Sampling*: The diffusion timestep $t$ is sampled from a uniform distribution. This discrete sampling process is non-differentiable and not suitable for policy gradient optimization.

- *Mode Selection*: The logits $\mathbf{l} = [l_1, l_2, \ldots, l_K]^\top$ are produced by the learnable decoder $f_\theta$ and represent the context-dependent confidence scores of the model for each anchor. The resulting categorical distribution:

$$P(\text{mode} = k|\mathbf{q}) = \sigma(\mathbf{l})_k = \frac{\exp(l_k)}{\sum_{j=1}^K \exp(l_j)} \tag{10}$$

is both differentiable and optimizable, as the logits $\mathbf{l}$ are functions of the model parameters $\theta$.

Based on this analysis, we define the surrogate probability for trajectory $\mathbf{o}_k$ (generated from the $k$-th anchor) simply as follows, intentionally excluding the fixed prior distribution as they cannot be optimized through reward signals:

$$\pi_\theta(\mathbf{o}_k|\mathbf{q}) = P(\text{mode} = k|\mathbf{q}) = \sigma(\mathbf{l})_k \tag{11}$$

With the surrogate probability $\pi_\theta$ properly defined, the GRPO integration becomes straightforward. For each input scenario $q$, DiffusionDrive naturally produces a group of $G = N_{\text{anchor}}$ trajectory proposals $o_i, i = 1, ..., G$, each generated from a distinct anchor trajectory. We directly use this set of trajectories as the group of "responses" required by GRPO. The probability $\pi_\theta(o_i|q)$ for each trajectory is given by the surrogate probability distribution defined in Eq. (4).

