# OpenReview forum: "READ: End-to-End Autonomous Driving Made Safer with Efficient Reinforcement Learning"
_ICLR.cc/2026/Conference — ICLR 2026 Conference Withdrawn Submission_

### Official Review · Reviewer_g2TG · 2025-10-24

**Soundness:** 2
**Presentation:** 2
**Contribution:** 2
**Rating:** 4
**Confidence:** 4

**Summary:**

This paper proposes ReAD, a reinforcement learning based framework for end-to-end autonomous driving. The authors introduce a surrogate probability to enable GRPO policy optimization in a continuous trajectory space, and design a reward decomposition strategy to provide clear and disentangled learning signals. ReAD can be efficiently applied to existing multimodal models such as DiffusionDrive, achieving improved safety and progress with minimal training cost. Experiments on NavSim and nuScenes demonstrate meaningful improvements, especially in collision reduction.

**Strengths:**

The paper addresses a recognized limitation of imitation-based planning. The reward decomposition strategy is novel in the context of end-to-end driving and demonstrates clear effectiveness through ablations. The method requires no architectural change and only lightweight fine-tuning, suggesting strong practical value in industry. Improvements in collision reduction are compelling and show that RL can recalibrate multimodal distributions beneficially.

**Weaknesses:**

1. The paper mainly formalizes a general RL setup for driving, but lacks deeper analysis of why this formulation works, how it differs fundamentally from prior RL attempts, and how design choices relate to driving-specific policy optimization challenges.

1. ReAD(DiffusionDrive) only uses anchor selection logits while discarding the denoising generation process, which is the major stochastic component.

1. It also relies on a model’s internal mode-selection logits (e.g., anchor trajectories in DiffusionDrive). It’s unclear if ReAD applies to end-to-end models without such discrete, learnable confidence mechanisms, e.g. pure diffusion/flow matching models.

2. The reward decomposition works well but uses manually tuned weights (Table 1). The paper lacks details on how these weights were chosen, analysis on cooperative and conflicting dynamics among objectives, and why optimization directions differ between independent and aggregated rewards.

3. ReAD is not compared directly with recent RL-based driving methods (e.g., AlphaDrive, Drive-R1), making it hard to assess its relative advancement.

**Questions:**

Same as weaknesses

---

### Official Review · Reviewer_qnEs · 2025-10-30

**Soundness:** 3
**Presentation:** 3
**Contribution:** 3
**Rating:** 4
**Confidence:** 4

**Summary:**

The paper proposes ReAD, a lightweight RL fine-tuning scheme for end-to-end driving. Two ingredients:
(1) a surrogate probability for continuous-trajectory planners (derived from internal, differentiable confidence logits), enabling GRPO-style policy updates without discretizing trajectories; and
(2) reward decomposition: instead of one composite driving reward (e.g., PDMScore), compute separate GRPO losses for semantically aligned components (NC, DAC, TTC, EP, Comfort) and optimize a weighted sum. Empirically, ReAD fine-tunes DiffusionDrive and improves NavSim closed-loop PDMS (+1.1) and nuScenes open-loop collision rate

**Strengths:**

Pros:

1. Practical RL integration for E2E planners. Avoids heavy critics and awkward discretizations. The surrogate log-prob grounded in mode-selection logits is simple and implementable across many multi-modal planners.
2. Reward decomposition is well-motivated and effective. Clear ablations show monolithic PDMScore gives little gain, single-objective EP harms safety, while decomposed losses improve multiple axes at once.

**Weaknesses:**

Cons:
1. The paper lacks sufficient innovation. The concept of reward decomposition, splitting the overall objective into individual metrics, is not a particularly novel contribution within the field of Reinforcement Learning.
2. The experiments are incomplete. The paper fails to provide a clear justification for the specific weight assignments for the decomposed PDMScore components. The choice of weights seems to be validated only through ablation studies of different combinations rather than a principled basis.
3. The baseline PDMScore for DiffusionDrive is cited as 88.1 (whereas this paper reports 87.7). Furthermore, the practical utility of the method requires further validation. The paper does not include a comparison against state-of-the-art (SOTA) algorithms, such as GoalFlow (which achieves a PDMScore of 90.3). The final score of 88.8 is relatively poor on the NavSim benchmark leaderboard, failing to demonstrate the superiority of the proposed method.
4. The Appendix defines π(o|q) solely via mode-selection softmax (ignoring diffusion noise and timestep sampling). That pushes all policy credit through selection logits, not through the denoising/regression pathway. The paper claims the surrogate should “engage a substantial portion” of parameters, but the final instantiation may bypass large parts of the generator. This gap warrants stronger justification or analysis.
5. Limited evaluation scope. NavSim is non-reactive; nuScenes evaluation is open-loop. Furthermore, owing to distributional fragmentation and sampling bias in nuScenes, end-to-end evaluations on this dataset do not constitute credible validation. Safety claims would be stronger with reactive closed-loop tests (e.g., interactive sim or traffic agents responding to ego), or at least with intervention-style metrics.
6. No latency / real-time analysis; RL regularization could skew confidence calibration and affect deployment rate control.
7. Hyperparameter sensitivity / weight tuning. The decomposition relies on hand-picked wNC, wDAC, wTTC, wEP, wComf. The paper tunes these but does not quantify sensitivity or overfitting risks (e.g., by multiple seeds, cross-scenario validation).

**Questions:**

1. I am curious how robust the gains are to noise-free vs noisy denoising at inference: if you inject diffusion noise at test time, do we still observe improvements, given the surrogate omits noise terms?
2. The group size G equals the number of anchors (20 or 40). Have you tried multiple samples per anchor (varying diffusion randomness) to enrich GRPO’s within-anchor diversity?
3. Does the KL to πref ever dominate learning (early collapse)? Show KL/advantage traces and any temperature/scale calibration applied to logits.
4. Any evidence of reward hacking (e.g., EP increases via uncomfortable accelerations despite Comfort term)? Provide qualitative rollouts.

---

### Official Review · Reviewer_zPiE · 2025-11-03

**Soundness:** 2
**Presentation:** 2
**Contribution:** 2
**Rating:** 2
**Confidence:** 5

**Summary:**

READ proposes a lightweight reinforcement-learning fine-tuning framework for end-to-end autonomous driving that (i) introduces a surrogate, differentiable probability over multimodal trajectory generation to enable GRPO in a continuous action space and (ii) decomposes a composite driving reward into semantically aligned components (NC, DAC, TTC, EP, Comfort) and optimizes them as separate policy-gradient losses to improve credit assignment and convergence. Empirically, fine-tuning a DiffusionDrive planner for only a couple of epochs boosts PDMScore on NavSim, and achieves better collision rate on nuScenes without increasing L2 error. Further ablations show monolithic rewards underperform and that the decomposition and richer anchor sets drive the gains.

**Strengths:**

- **Clarity.** Well-written with a tight problem setup and intuitive figures/notation.
- **Reproducibility.** Detailed training/evaluation settings and ablations enable faithful replication.
- **Robust Gains.** Consistent improvements across **nuScenes** and **NAVSIM** on safety, efficiency, and composite metrics with short fine-tuning.

**Weaknesses:**

- Missing analysis in multi-objective RL. The current formualtion includes reward decomposition and joint optimization of different GRPO losses. This potentially poses similar challenge in previous MORL and gradient surgery literautres. However, these related works are completely omitted. And the gradient conflicting issue is not discussed and properly approached in the paper. The authors just give a final selection of hyperparameter weights in Table 1 without good insights from ML background. This brings down the technical soundness of the paper.
- Recent works on VLA [3, 4, 5] also utilize GRPO-like RL algorithm to optimize the performance on NAVSIM benchmark also nuScenes. Diffusion-based planning [6] can also utilize reward guidance to steer the denoising process of diffusion-based planner. Yet the paper only compares with one contemporary E2E driving baseline, i.e. DiffusionDrive. More comparison with similar baselines using inference-time reward guidance, or the baselines using GRPO on foundation models will further demonstrate how effective the reward trick could be and help the audience better fairly present the core contribution of the paper.
- Minor: Typos in Gradient-Policy-Loss. Should be Policy Gradient loss in GRPO?

> [1] Yang, Runzhe, Xingyuan Sun, and Karthik Narasimhan. "A generalized algorithm for multi-objective reinforcement learning and policy adaptation." NeurIPS 2019.
>
> [2] Yu, Tianhe, et al. "Gradient surgery for multi-task learning." NeurIPS 2020.
>
> [3] Zhou, Zewei, et al. "AutoVLA: A Vision-Language-Action Model for End-to-End Autonomous Driving with Adaptive Reasoning and Reinforcement Fine-Tuning.", 2025.
>
> [4] Jiang, Bo, et al. "Alphadrive: Unleashing the power of vlms in autonomous driving via reinforcement learning and reasoning." arXiv 2025.
>
> [5] Li, Yongkang, et al. "Recogdrive: A reinforced cognitive framework for end-to-end autonomous driving." *arXiv 2025.
>
> [6] Zheng, Yinan, et al. "Diffusion-based planning for autonomous driving with flexible guidance." ICLR 2025.

**Questions:**

- GRPO is primarily designed for LLMs setting, where rollout and value critics are expensive. What stops the author from using other on-policy RL algorithm like PPO?
- Do the authors observe any forgetting issues and objective conflict when jointly optimizing the multiple objectives? Besides the weighted sum over the GRPO loss, have they attempted using alternative solutions?
- Will it be a good idea to use some parameter-efficient fine-tuning for the RL algorithm on different reward objectives, to avoid overfitting in a more principled manner?

---

### Note · Authors · 2025-11-14

I have read and agree with the venue's withdrawal policy on behalf of myself and my co-authors.